:ᐧ҉: PLOS | ONE

# Regulation of nitrogen availability results in changes in grain protein content and grain storage subproteomes in barley (*Hordeum vulgare* L.)

**Baojian Guo[☯], Dongfang Li[☯], Sen Lin[☯], Ying Li, Shuang Wang, Chao Lv, Rugen Xu[ID]***

Jiangsu Key Laboratory of Crop Genetics and Physiology/Co-Innovation Center for Modern Production Technology of Grain Crops, Key Laboratory of Plant Functional Genomics of the Ministry of Education, Barley Research Institution of Yangzhou University, Yangzhou University, Yangzhou, China

[☯] These authors contributed equally to this work.
* rgxu@yzu.edu.cn

**Data Availability Statement:** All relevant data are within the manuscript and its Supporting Information files.

## Abstract

Barley grain protein content (GPC) is an important quality factor that determines grain end-use value. The synthesis and accumulation of grain protein is highly dependent on the availability of nitrogen fertilizer, and it is important to understand the underlying control mechanisms of this. In the current study, the GPC and protein composition of mature grain seeds from Yangsimai 3 and Naso Nijo barley cultivars were analyzed. Grain storage subproteomes (albumin, glubulin, hordein and glutelin) were compared in the cultivars grown in both low and high nitrogen level conditions. The GPC of mature grain was significantly higher in Yangsimai 3 than Naso Nijo following nitrogen treatment. Albumin, hordein and glutelin content were increased in Yangsimai, while only hordein content was increased in Naso Nijo. Large-scale analysis of the grain storage subproteome revealed 152 differentially expressed protein spots on 2-DE gels with a pH range of 3–10. Among these, 42 and 66 protein spots were successfully identified by tandem mass spectrometry in Yangsimai 3 and Naso Nijo grown in low and high nitrogen conditions. The identified proteins were further grouped into thirteen categories according to their biological functions. This detailed analysis of grain subproteomes provides information on how barley GPC may be controlled by nitrogen supply.

## Introduction

Barley (*Hordeum vulgare* L.) was one of the first crops to be domesticated and was a founder crop in planting areas throughout the world [1–2]. Protein is one of most important nutrient components of barley grain, and it has long been suggested that grain protein content (GPC) can be increased with an appropriate amount of nitrogen fertilizer [3–4]. According to their solubility, seed storage proteins can be classified as albumins (water soluble), globulins (alkaline soluble, water insoluble), hordeins (alcohol soluble), or glutelins (alkaline soluble, but

**Funding:** This research was supported from the National Natural Science Foundation of China (31771771, 31401370 and 31571648), Natural Science Foundation of the Jiangsu Higher Education Institutions of China (17KJB210006), National Barley and Highland Barley Industrial Technology Specially Constructive Foundation of China (CARS-05), and a Project Funded by the Priority Academic Program Development of Jiangsu Higher Education Institutions. The funders had no role in study design, data collection and analysis, decision to publish, or preparation of the manuscript.

**Competing interests:** The authors have declared that no competing interests exist.

water, alcohol and saline insoluble) [5]. Hordeins and glutelins are the major storage proteins in grain seeds, while albumins and globulins mainly comprise enzymes or enzyme inhibitors involved in cell metabolism and development [6–7]. Until now, it was unclear which protein subunit was directly altered by changes to nitrogen levels.

Proteomics has been employed to dissect the genetic basis of GPC and quality [8–13]. For example, using comparative proteomics, Görg et al. [8–9] identified a 19 kDa hordein-like polypeptide that plays a role in determining malting quality in barley cultivars, as it was quickly degraded in low malting grade grain. In addition, two-dimensional gels combined with tandem mass spectrometry have been used to investigate the processes of grain filling, maturation and germination in barley seeds [11]. Accumulation of proteins during cereal grain maturation has been shown to be relative to changes at the transcriptional and proteomic levels and differs with nitrogen level [14–15]. Changes in the expression of transport and metabolism genes caused by changes to nitrogen and sulphur supplies led to altered concentrations of several free amino acids. These amino acids seem to be essential in determining expression and accumulation of grain storage proteins in wheat [14]. In addition, several differentially expressed proteins potentially related to grain storage protein accumulation in diploid wheat were identified as central actors in the response to nitrogen levels using proteomics [16].

In the present study, we report a comparative proteomics analysis of two cultivars (Yangsimai 3 and Naso Nijo) in high and low nitrogen conditions using two-dimensional gel electrophoresis (2-DE) and tandem Mass Spectrometry (MS). The main objectives were: (1) to investigate the difference in GPC and protein composition between low and high nitrogen-treated plants; (2) to obtain comparative information on the subproteomes of feed barley and malting barley under low and high nitrogen conditions; (3) to identify changes in protein composition and candidate proteins that increase GPC under high nitrogen conditions.

## Materials and methods

### Plant material and growth conditions

Two barley cultivars, Yangsimai 3 and Naso Nijo, were used in this study. Yangsimai 3 is a Chinese cultivar of feed barley, and is two-rowed, with a high grain protein content. Naso Nijo is a Japanese two-rowed malting barley cultivar with a low GPC [13]. Both barley cultivars were planted at the Yangzhou University Experimental Farm in autumn of 2014 in soil with a total N of 1.14g/Kg. Cultivars were grown under low (0 kg N/ha) and high (225 kg N/ha) nitrogen conditions. Urea was given as a base nitrogen application just before sowing. Forty seeds of each cultivar were planted 3 cm apart with 25 cm between rows. Mature seeds were harvested from the middle region of the main spikelet, and then seeds were dried to a consistent level, and stored at -20°C for seed total protein extraction and protein composition (including albumin, globulin, hordein and glutelin) analysis. A total of 100 seeds per genotype used for protein extraction and protein content measurement for each replication. Three biological replicates were performed in the present study.

### Extraction of proteins and protein composition content measurement

Mature grains were ground in a Cyclotec 1093 sample mill (Hoganas City, Sweden) and sieved through a 0.5 mm screen. Proteins were extracted from samples (1 g) according to the methods described by Shewry *et al.* [17], with some modifications. Albumin extraction was carried out three times for 30 min each with 2 mL deionized water: supernatant was collected by centrifugation at 1,000 g for 10 min, and was used for albumin protein content measurement. The precipitate was used to determine globulin content and was dissolved in 0.5 M sodium chloride.

This step was repeated three times, whereby the supernatant was collected following centrifugation at 1,000 g for 10 min, washed once with deionized water and the supernatant containing sodium chloride was removed. Similarly, hordein extraction was carried out six times for 1 h each with 55% (v/v) propan-2-ol, 1% (v/v) acetic acid and 1% (v/v) DTT (15–30 ml) at 60°C. Samples were centrifuged at 12,000 g for 10 min to permanently disrupt the disulphide bonds within and between hordeins, pure 4- vinyl-pyridine was added and mixtures (v/v, 1.4%) were shaken for 30 min at 60°C [18]. The precipitate was washed once with deionized water and the supernatant containing the hordein extraction buffer was removed. The precipitate was resuspended in 0.05 M sodium hydroxide and centrifuged at 12 000 g for 10 min three times. The supernatants containing glutelin were collected. Extracted proteins were used for protein content measurement and protein powder preparation. The nitrogen content and protein composition of whole grain samples were quantified according to the Kjeldahl method using a FOSS Kjeltec $^{TM}$ 2300 analyzer unit (Foss Analytical AB, Sweden) [19]. Protein content (grain protein content, albumin content, globulin content, hordein content and gluten content) was calculated using the following formula: Protein content = Nitrogen content×5.83×100% [20]. Statistical analysis of the differences in aerial part traits between cultivars was performed using Student's $t$-tests.

The proteins (albumin, globulin, hordein and glutelin) were precipitated with 10% TCA containing 0.07% DTT to remove the extraction buffer, which interferes with isoelectric focusing (IEF). One volume of sample was added to 4 volumes of cold 10% TCA solution. After 2 h incubation at -20°C, the extracts were centrifuged at 18,000 g for 30 min at 4°C and the supernatant was discarded. Protein pellets were resuspended with cold 80% acetone containing 0.07% DTT and incubated for 1 h at -20°C before centrifugation at 18 000 g for 15 min at 4°C. This step was repeated five times and the protein pellet was freeze-dried under a vacuum. Protein pellets (albumin, globulin, hordein and glutelin) were solubilized and incubated in a protein buffer (7 M urea, 2 M thiourea, 2% CHAPS (powder to solution, w/v), 0.5% IPG buffer (v/v) (pH 3–10) (Fairfield City, USA) and 36 mM DTT (5.6 mg/mL)) at room temperature for 1 h and vortexed every 10 min. The mixture was then centrifuged (20 000 g) for 15 min, and the supernatant was collected. Protein concentration was determined by Bradford assay [21] with bovine serum albumin (BSA) as a standard.

## Two-Dimensional gel electrophoresis and image analysis

Seed protein extract (200 μg) was loaded onto a GE Healthcare 18 cm IPG strip with a linear gradient of pH 3–10 during overnight strip rehydration. IEF was conducted using IPGPhorII (Fairfield City, USA) at 20°C for a total of 65 kVh. Equilibration of the strips was performed immediately with 10 mL of two types of SDS equilibration buffer for 15 min each. Buffer 1 contained 1.5 M Tris-HCl (pH 8.8), 6 M urea, 30% glycerol, 2% SDS, and 1% DTT, and buffer 2 contained 1.5 M Tris-HCl (pH 8.8), 6 M urea, 30% glycerol, 2% SDS, and 2.5% iodoacetamide. Second dimension SDS-PAGE gels (12.5% linear gradient) were run on an Ettan DALT-six (Fairfield City, USA) for 0.5 h at 2.5 W per gel, then at 12 W per gel until the dye front reached the end of the gel. Upon electrophoresis, the protein spots were stained with silver nitrate according to the instructions of the PlusOne™ Silver Staining Kit for proteins (Fairfield City, USA), which offered improved compatibility with subsequent mass spectrometric analysis. Briefly, gels were fixed in 40% ethanol and 10% acetic acid for 30 min, and then sensitized with 30% ethanol, 0.2% sodium thiosulfate (w/v) and 6.8% sodium acetate (w/v) for 30 min. Gels were then rinsed with distilled water for five minutes three times, then incubated in silver nitrate (2.5 g/L) for 20 min. Incubated gels were rinsed with distilled water and developed in a sodium carbonate solution (25 g/L) with formaldehyde (37%, w/v) added (300 μL/L) before

use. Development was stopped with 1.46% EDTA-Na$_2$•2H$_2$O (w/v), and gels were stored in distilled water until they could be processed and reproducible spots were removed from them. Gel images were acquired using Labscan (Fairfield City, USA). Image analysis was carried out with Imagemaster 2D Platinum Software Version 7.0 (Fairfield City, USA). Three biological replicates of silver stained gels showed high reproducibility (>95%) when compared using Imagemaster 2D Platinum Software 7.0. Spot detection was performed automatically by the software with the parameters smooth, minimum area and saliency set to 2, 15 and 8, respectively, followed by manual spot editing, such as spot deletion, spot splitting and merging. All gels were matched to the reference gel in automated mode with Imagemaster 2D Platinum Software v7.0. The volume of each spot from three replicate gels was normalized and quantified against total spot volume using Imagemaster 2D Platinum Software 7.0. Sequential k-nearest neighbor methods was used to impute missing values. Changes in the normalized spot volumes between experimental and control images were evaluated with a mixed linear mode. The spot number and normalized spot volume data were formatted in Excel. Protein expression in Yangsimai 3 and Naso Nijo was compared using Student $t$-tests, and only those protein spots with fold changes greater than 1.5 and $p<0.05$ were considered differentially expressed protein spots.

When comparing the patterns of protein expression under low and high nitrogen levels, both quantitative and qualitative differences were observed. The quantitative differences can be grouped into two categories: up-regulated or down-regulated protein spots in Yangsimai 3 (UY or DY) and Naso Nijo (UN or DN). The qualitative differences can also be grouped into two categories: specific expressed protein spots in Yangsimai 3 (SEPSY) and specific expressed protein spots in Naso Nijo (SEPSN). Student's $t$-tests ($p<0.05$) were used to determine significant differences in relative abundances of protein spot features between Yangsimai 3 and Naso Nijo. Spots with reproducible and significant variations, at least 1.5-fold up-regulated or down-regulated, were considered quantitative differentially expressed proteins.

## In-gel digestion of proteins

Protein spots were excised manually and transferred to 1.5 mL microcentrifuge tubes, and proteins with low abundances were removed from all the replicate gels to be pooled and digested in a single tube. Protein spots were destained twice with 30 mM potassium ferricyanide and 100 mM sodium thiosulfate, and then rinsed with 25 mM ammonium bicarbonate in 50% acetonitrile. Protein spots were dehydrated with 100% acetonitrile, dried under vacuum, and 10 µL trypsin (10 ng/µL) was added, and imbibed for 40 min on ice. Protein spots were then covered with 25 µL 25 mM ammonium bicarbonate and incubated for 16 h at 37˚C. The peptides were eluted using 30 µL 0.1% TFA, shaken for 10 min, and the digestion solution was transferred to a new 1.5 mL tube before the protein spots were eluted using 70% v/v acetonitrile and 0.1% v/v trifluoroacetic acid twice. The digestion solution was then transferred once more to a new 1.5 mL tube, incorporating the digestion solution, and freeze-dried for 2 h to condense the volume to 10 µL before storage at -80˚C.

## Identification of proteins by mass spectrometry

The digestion solution was spotted on an MALDI target plate (1.0 µL) twice and the recrystallized CHCA matrix was dissolved in 0.1% TFA/70% ACN (0.5 µL). A Mass Standards Kit for Calibration of SCIEX MALDI-TOF Instrument (Foster City, USA) was used for mass assignment. Each sample spot was desalted with 0.01% TFA, and completely dried. Protein identification was conducted using an SCIEX MALDI TOF-TOF™ 5800 analyzer equipped with neodymium. For the MS mode, peptide mass maps were acquired in positive reflection mode,

and the 800–4,000 m/z mass range was used with 4,000 laser shots per spectrum. A maximum of 20 precursors per spot with a minimum S/N ratio of 20 were selected for MS/MS analysis in 2 kV positive modes. The contaminant m/z peaks originating from trypsin auto-digestion, or matrix, were excluded from MS/MS analysis.

Combined MS and MS/MS results were analyzed using ProteinPilot software (Foster City, USA), and the results were searched using MASCOT software (http://www.matrixscience.com/). Matches to protein sequences from the Viridiplantae taxon (other green plants) in the NCBInr database (updated 6 June 2014) were considered acceptable if: 1) A MOWSE score was obtained from MASCOT, which rates scores as significant if they are above the 95% significance threshold ($p < 0.05$); 2) At least two different predicted peptide masses matched the observed masses for an identification to be considered valid; 3) The coverage of protein sequences by the matching peptides was higher than 5%; 4) A peptide mass tolerance of ±0.15 Da was achieved; 5) A parent ion mass tolerance of 0.2 Da was achieved; and 6) Acetylation of the N-terminus, cysteine as carboxylamidomethyl cysteine, pyroglu formation of N-terminal Gln, and methionine in an oxidized form were set as possible modifications. To understand their function, the identified proteins were classified using the MapMan ontology defined by Guo et al. [13].

## Results

### Effects of nitrogen supply on grain protein content and protein composition

In the present study, total GPC in low nitrogen level plants was 12.22% and 12.13% in Naso Nijo and Yangsimai 3, respectively, and 14.02% and 14.72%, respectively in high nitrogen level plants. The difference between low nitrogen level and high nitrogen level PGCs was statistically significant ($p < 0.05$) (Fig 1A). In addition, Yangsimai 3 had a higher GPC than Naso Nijo under high nitrogen conditions ($p < 0.05$) (Fig 1A). Hordeins are the major storage proteins in barley. The hordein content of both Yangsimai 3 and Naso Nijo under high nitrogen conditions was significantly higher than at low nitrogen conditions, at 34.72% and 17.10% (p < 0.01), respectively (Fig 1B). A significant difference was also observed between Yangsimai 3 and Naso Nijo under high nitrogen conditions. Albumin and gluten contents increased with nitrogen supply in Yangsimai 3, however, albumin content was inhibited by high nitrogen in Naso Nijo (Fig 1B). Thus, the increased hordein content determined the GPC under high nitrogen conditions.

### Construction of subproteomes of albumin, globulin, hordein and glutelin from two cultivars under low and high nitrogen conditions

To construct a 2-DE map of barley grain proteins, albumin, globulin, hordein and glutelin were extracted from Yangsimai 3 and Naso Nijo and separated by 2-DE with three biological replicates. At a linear gradient of pH 3–10, a total of 175, 259, 22 and 111 protein spots were observed on Yangsimai 3 2-DE gels with the protein expression profiles of albumin, globulin, hordein and glutelin, respectively, while 144, 286, 22 and 152 protein spots were detected on Naso Nijo 2-DE gels (Fig 2). In total, 152 reproducible differentially expressed protein spots were detected in both barley varieties, among which 68 (68/567, 11.99%) and 84 (84/604, 13.91%) protein spots were found to have different patterns of expression in Yangsimai 3 and Naso Nijo between high and low nitrogen conditions (student's *t*-test at p < 5%) (Table 1).

When analyzing different patterns of expression in low nitrogen level and high nitrogen treated plants, both quantitative and qualitative differences were observed. Student's *t*-tests

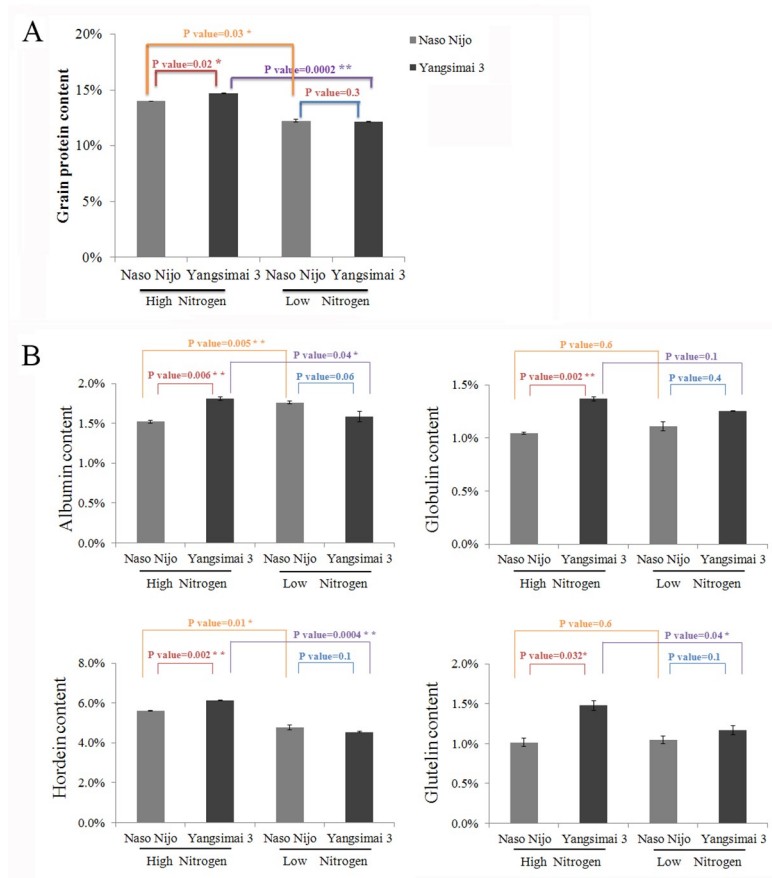

**Fig 1. The total grain protein contents and protein fractions in mature grains of two barley cultivars grown under high and low nitrogen level conditions.** A, total grain protein content of mature grain. B, protein compositions (albumin, globulin, hordein and glutelin) of mature grain.

were used to calculate significant differences in the relative abundance of protein spots. A total of 68 and 84 protein spots were identified as differentially expressed proteins in Yangsimai 3 and Naso Nijo, respectively. The qualitative differences were grouped into two categories: SEPSY, which had 19 entries, and SEPSN, which had 21 entries (Table 1). The quantitative differences were grouped into up-regulated or down-regulated categories in Yangsimai 3 (UY with 22 entries or DY with 27 entries) and Naso Nijo (UN with 31 entries or DN with 32 entries) (Fig 3, Table 1). A total of 77 and 54 differentially expressed protein spots were observed in the globulin and albumin expression profiles, respectively, but only 10 and 11 were detected in hordein and glutelin expression profiles, respectively (Table 1).

## Identification of differentially expressed protein spots

All differentially expressed protein spots between high and low nitrogen level conditions in both varieties (Yangsimai 3 and Naso Nijo) were excised from representative 2-DE gels for identification. In total, 108 differentially expressed protein spots were successfully identified by tandem mass spectrometry, corresponding to 64 unique proteins (Fig 2, Table 1, S1 Table). These identified proteins were further grouped into thirteen categories according to their biological functions, and the category with the most identified proteins was stress (11/65 proteins, 17%), followed by protein degradation and posttranslational modification, TCA, redox,

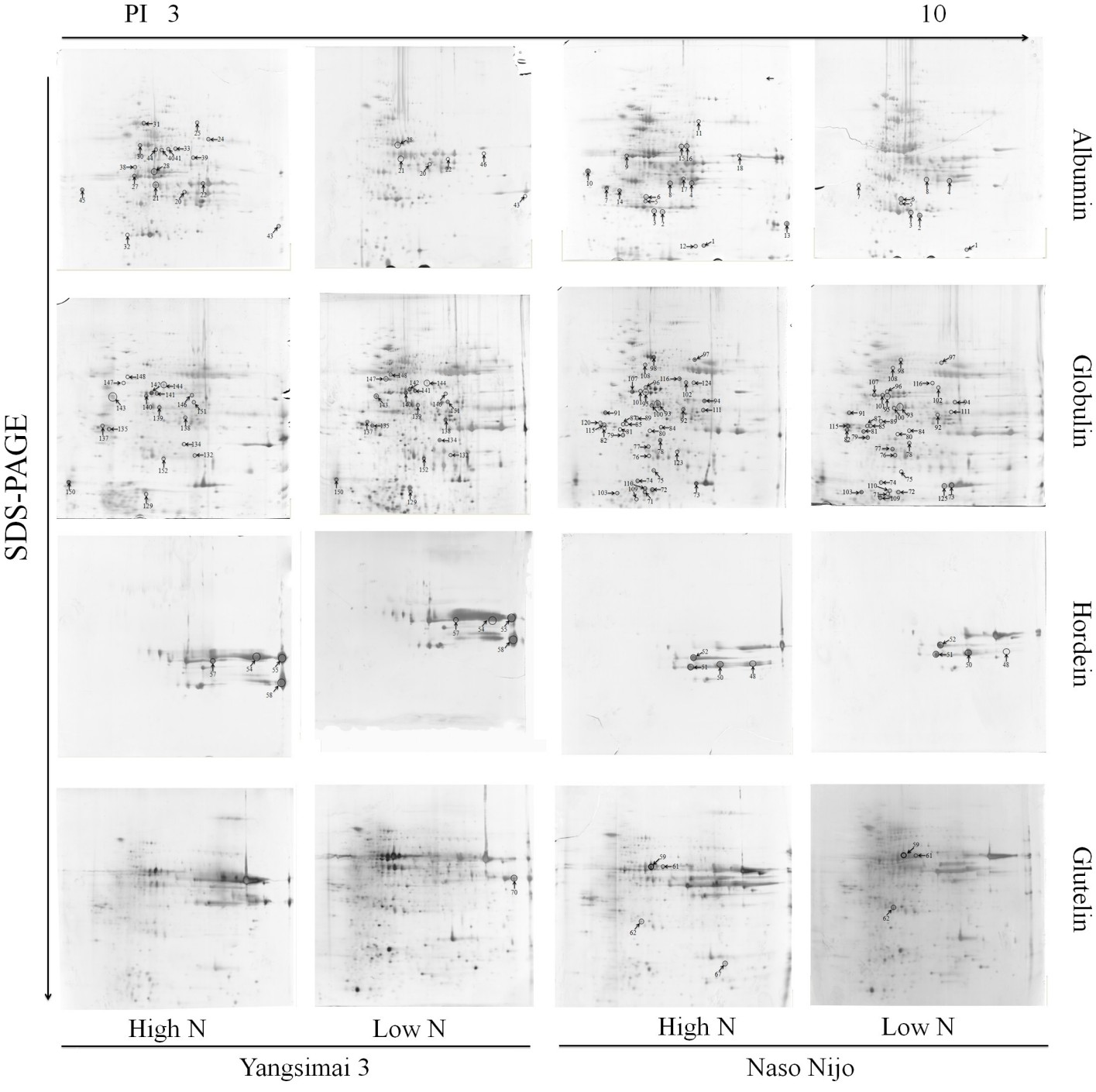

**Fig 2. Protein expression profiling analysis of two barley cultivars with contrasting grain protein contents between high and low nitrogen levels.**

glycolysis, development, and metabolism categories, which contained 10, 6, 5, 4, 4, and 4 proteins, respectively. A very small number of proteins functioned in photosynthesis (2 entries), RNA function (2 entries), signaling (2 entries), biodegradation (1 entries) and fermentation (1 entries). The remaining proteins with unknown functions were categorized as other function (12 entries) (Fig 3).

**Table 1. Summary of the number of differentially expressed spots between low and high nitrogen condition.**

| Expression profiling | Differentially expressed protein spots | | | | | | Total |
|---|---|---|---|---|---|---|---|
| | Yangsimai 3 | | | Naso Nijo | | | |
| | SEPSY | UY | DY | SEPSN | UN | DN | |
| Albumin | 12 (12) | 6 (4) | 5(3) | 11 (9) | 9 (3) | 11 (6) | 54 (37) |
| Globulin | 6 | 10 (1) | 17 (17) | 9 (5) | 18 (18) | 17 (17) | 77 (58) |
| Hordein | 0 | 3 (2) | 3 (2) | 0 | 3 (3) | 1 (1) | 10 (8) |
| Glutelin | 1 (1) | 3 (0) | 2 (0) | 1 (1) | 1 (1) | 3 (2) | 11 (5) |
| Total | 19 (13) | 22 (7) | 27 (22) | 21 (15) | 31 (25) | 32 (26) | 152 (108) |

Note, the digit in brackets indicate differentially expressed proteins spots were identified by tandem MS.

Further analysis revealed that these 64 unique proteins were derived from 49 different genes or gene families, and 62 protein spots corresponded to 20 protein isoforms (S1 Table). These isoforms were identified by the same gene ID, though they differed significantly with respect to their pIs and Mr. The number of isoforms for each protein ranged from 2 to 8. For example, serpin-Z4 (gi|1310677) and serpin-Z7 were identified in eight (spots 28, 38, 41, 96, 141, 142, 59 and 61) and seven protein spots (spot 9, 40, 95, 101, 107, 108 and 140), respectively, while spots 82 and 135 were identified as a 14-3-3 protein. In addition, some proteins had the same protein name with different gene IDs (S1 Table). For example, three protein spots were identified as 14-3-3 proteins, but the gene IDs were gi|326493920 (spot 82 and 135) and gi|257664738 (spot 137).

## Discussion

### The grain protein content of Yangsimai 3 and Naso Nijo displayed different patterns in response to nitrogen treatment

Nitrogen is one of the most important soil nutrients for ensuring both high grain yield and grain quality, and increasing yield and protein content are major objectives for cereal crop breeding programs [22]. Generally, a high GPC in barely grains is required for human food

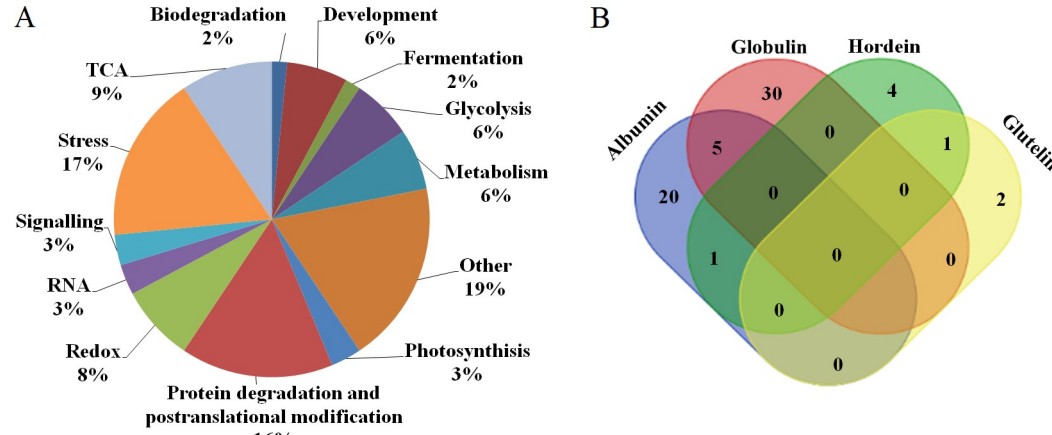

**Fig 3. Functional category and Venn diagram of differentially expressed proteins.** A, functional category of differentially expressed proteins; B, Venn diagrams showing the number of differentially expressed proteins common to ('overlap' genes) and specifically expressed in the four subproteomes. Numbers in a single-shaded region indicate subproteome-specific proteins, while those in a double-shaded region show the overlap proteins.

and animal feed, while low GPC is desirable for malting and brewing industries. The form and concentration of nitrogen (N) in soil is a major factor in determining grain quality and crop yield [23]. Remarkably, barley grain protein content is significantly affected by different nitrogen levels and cultivars; increasing nitrogen fertilizer application increases the hordein protein content (lysine-poor storage protein) with the increase in total grain protein content, but there are only small increases in the total amounts of the other more lysine-rich proteins (salt soluble proteins and glutelin residue proteins) [6, 24]. In the current study, grain protein content of the two cultivars increased with the level of nitrogen and displayed significant differences under high nitrogen levels. Albumin, globulin, hordein and glutelin content increased in Yangsimai 3, but only hordein content increased in Naso Nijo under high nitrogen treatment. Hordein content increased by 1.58% in Yangsimai 3 compared to 0.82% in Naso Nijo. Therefore, the hordein fraction in barley grains was more sensitive to nitrogen and increased as the nitrogen level increased, which is the most stable factor for high GPC in feed barley.

## Differentially expressed proteins contributed to the difference in grain protein content with nitrogen treatment

Changes in proteome under low and high nitrogen conditions have also been observed; with differentially expressed proteins involved in metabolism, stress, glycolysis, tricarboxylic acid cycle, protein degradation, carbon fixation and other functions [25–28]. Recently, proteomics has been employed to investigate the grain response to nitrogen supply, and enzymes or enzyme inhibitors involved in cell metabolism and development were detected in albumin and globulin subproteomes [16]. In the present study, a large number of differentially expressed proteins were identified in albumin (37 proteins) and globulin (58 proteins) expression profiles, which had functional roles in twelve categories (S1 Table). For example, five differentially expressed protein spots (spots 24, 92, 94, 111 and 151) were identified as glyceraldehyde-3-phosphate dehydrogenase (GAPDH), of which spot 24 derived from the albumin expression profile and was up-regulated in Yangsimai 3 under high nitrogen conditions. Others were detected in the globulin expression profile and displayed various expression patterns. GAPDH is considered a classical glycolytic protein and exhibits distinct activities as a multidimensional protein, such as iron metabolism, membrane trafficking, histone biosynthesis, the maintenance of DNA integrity and receptor mediated cell signaling [29]. Its expression is altered by low and high N levels combined with different sulphur fertilization levels in wheat [27]. Serpin-Z4 and Z7 belong to the barley serine protease inhibitor (serpin) family, and both play important roles in improving beer foam-stability and malt filterability [30–31]. In our previous study, we identified multiple expression patterns in Yangsimai 3 and Naso Nijo [13]. In the albumin and globulin proteomes, a total of 7 and 8 Serpin protein spots were detected in Yangsimai 3 and Naso Nijo, respectively. This suggests that the expression of Serpin-Z4/Z7 was induced by different nitrogen levels. Albumin and globulin are thus two subproteomes involved in GSP synthesis in grain seeds that are affected by nitrogen levels [16].

The major endosperm storage proteins are alcohol soluble hordeins in barley, which comprise 30–50% of the total grain protein [32–33]. According to electrophoretic mobility and amino acid composition, hordeins are generally divided into three groups: sulphur-rich (B, γ-hordeins), sulphur-poor (C-hordeins) and high molecular weight (HMW, D-hordeins) hordeins [34]. The hordein fractions were affected by cultivars and environmental variation in barley grain, as well as interactions between them [35–36]. Remarkably, B-hordein content account for 70–90% of the total hordein content, which was increased with nitrogen levels, and changes in B-hordein content accounted for the largest proportion of hordein content changes [6]. In the present study, we demonstrated the expression profiles of hordein and

glutelin under low and high nitrogen conditions, and specific differentially expressed hordein proteins were identified. A total of eight protein spots representing five hordein proteins were differentially expressed in Yangsimai 3 and Naso Nijo, which could contribute to the difference in GPC of mature barley seeds at the protein level.

### Comparative analysis of grain protein content in barley using transcriptomic and proteomics

*HvNAM1*, which encodes a NAC transcription factor as a *TtNAM-B1* orthologous gene in barley, was found to be responsible for the GPC QTL on barley chromosome 6HS [37–39]. At the transcriptional level, the *HvNAM1* gene was only found to be highly expressed in developing grain (15 day after pollination) and senescing leaves (56 day after pollination), with only trace amounts in other tissues [40]. In the present study, a total of 63 unique proteins were distributed across seven barley chromosomes (S1 Table). Among which, thirty genes were highly expressed in developing grain (5 and/or 15 days after pollination) (S1 Table). Remarkably, the GPC of both cultivars increased with increased nitrogen application, and Yangsimai 3 was more sensitive to nitrogen than Naso Nijo. The hordein proteome is a potentially major contributor to the difference in GPC between Yangsimai 3 and Naso Nijo. For example, protein spots 55 and 58 were identified as B hordeins and predicted proteins with unknown functions. Both accumulated in Yangsimai 3 with a higher abundance than in Naso Nijo. Therefore, multi-omics analysis can analyze the molecular mechanisms of GPC and nitrogen regulation in barley grains.

### Conclusion

In the present study, the GPC of mature grain was significantly higher in Yangsimai 3 than in Naso Nijo following nitrogen treatment. Hordein content was higher in both cultivars under high nitrogen conditions. Subproteome analysis revealed 152 differentially expressed protein spots on 2-DE gels. A total of 42 and 66 protein spots were successfully identified by tandem mass spectrometry in Yangsimai 3 and Naso Nijo under different nitrogen levels. Remarkably, the variation in B hordein content in Yangsimai 3 could have contributed to the higher GPC in Yangsimai 3 compared to Naso Nijo under different nitrogen availability conditions. Therefore, storage protein accumulation leads to GPC variation in feed and malting barley.

### Supporting information

**S1 Table. Peptide information of differentially expressed proteins identified by tandem mass spectrometry.**
(XLSX)

### Author Contributions

**Data curation:** Ying Li, Chao Lv.

**Formal analysis:** Sen Lin, Chao Lv.

**Funding acquisition:** Baojian Guo, Rugen Xu.

**Investigation:** Dongfang Li.

**Methodology:** Ying Li.

**Resources:** Dongfang Li.

**Supervision:** Shuang Wang.

**Validation:** Dongfang Li.

**Writing – original draft:** Baojian Guo, Dongfang Li, Rugen Xu.

**Writing – review & editing:** Baojian Guo, Rugen Xu.

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
