## [Decision Letter · Decision Letter 0]

2 Sep 2019

[EXSCINDED]

PONE-D-19-19801

Regulation of nitrogen availability results in changes in grain protein content and grain storage subproteomes in barley (Hordeum vulgare L.)

PLOS ONE

Dear Dr Xu,

Thank you for submitting your manuscript to PLOS ONE. After careful consideration, we feel that it has merit but does not fully meet PLOS ONE’s publication criteria as it currently stands. Therefore, we invite you to submit a revised version of the manuscript that addresses the points raised during the review process.

Although the research provided valuable information to understand the mechanisms for N regulation, the results were more descriptive and lack conclusions. It would be useful to have a detail comparison with the previous results. For this kind of research, detail information of soil nutrients is a must in the paper. The English  also need be thoroughly  edited  

We would appreciate receiving your revised manuscript by 25 September 2019. To enhance the reproducibility of your results, we recommend that if applicable you deposit your laboratory protocols in protocols.io, where a protocol can be assigned its own identifier (DOI) such that it can be cited independently in the future. For instructions see: http://journals.plos.org/plosone/s/submission-guidelines#loc-laboratory-protocols

We look forward to receiving your revised manuscript.

Kind regards,

Chengdao Li, PhD

Academic Editor

PLOS ONE

Journal Requirements:

https://onlinelibrary.wiley.com/doi/full/10.1111/tpj.12881

In your revision ensure you cite all your sources (including your own works), and quote or rephrase any duplicated text outside the methods section. Further consideration is dependent on these concerns being addressed.

Additional Editor Comments (if provided):

The MS reports different protein expression under different nitrogen supply. The research provided valuable information to understand the mechanisms for N regulation. The results were more descriptive and lack conclusions. It would be useful to have a detail comparison with the previous results. For this kind of research, detail information of soil nutrients is a must in the paper. The English need be thoroughly edited

Reviewers' comments:

Reviewer's Responses to Questions

**Comments to the Author**

1. Is the manuscript technically sound, and do the data support the conclusions?

Reviewer #1: Yes

Reviewer #2: Yes

2. Has the statistical analysis been performed appropriately and rigorously? 

Reviewer #1: Yes

Reviewer #2: Yes

3. Have the authors made all data underlying the findings in their manuscript fully available?

Reviewer #1: Yes

Reviewer #2: Yes

4. Is the manuscript presented in an intelligible fashion and written in standard English?

Reviewer #1: No

Reviewer #2: Yes

5. Review Comments to the Author

Reviewer #1: The manuscript “Regulation of nitrogen availability results in changes in grain protein content and grain storage subproteomes in barley (Hordeum vulgare L.) ” by Xu et al. describes comparison proteomics analysis of barley grain grown in both low and high nitrogen level conditions. The differential expression proteins were identified by MS. This detailed analysis of grain subproteomes provides information on how barley GPC may be controlled by nitrogen supply. Thus, the manuscript need minor revised.

1. The English style should be revised by native English speaker.

2. P20 Line 382, the authors referred “A total of eight protein spots representing five hordein proteins were differentially expressed in Yangsimai 3 and Naso Nijo, which could contribute to the difference in GPC of mature barley seeds. Why? The authors should be further discuss.

3. The introduction should explain the aim of the study.

4. References should be in uniform format as required by Plos one publication.

Reviewer #2: The authors presented grain protein content and grain storage subproteomes in barley under different N supply. Data were well presented. Following are my comments to the MS:

1. The MS is a bit too descriptive and lacks major conclusion. The mechanisms underline the changes need to be discussed, i.e. many differentially expressed proteins were detected (the same as previously reported) but their roles in causing the difference between two varieties were not discussed.

2. Not sure if it's appropriate to compare the results with previously reported QTL.

3. can the authors provide soil test results (how much N in the soil before N application)?

4. Don't think they can use “biological" before replicates if their replicates are from a single experiment.

5. A conclusion paragraph is needed at the end of Discussion.

6. PLOS authors have the option to publish the peer review history of their article (what does this mean?). If published, this will include your full peer review and any attached files.

Reviewer #1: No

Reviewer #2: No

---

## [Author Response · Author response to Decision Letter 0]

23 Sep 2019

Responses to reviewer 1

Comment 1. The English style should be revised by native English speaker.

Response 1: According to the reviewer’s suggestion, a native speak was invited to revise the manuscript.

Comment 2. P20 Line 382, the authors referred “A total of eight protein spots representing five hordein proteins were differentially expressed in Yangsimai 3 and Naso Nijo, which could contribute to the difference in GPC of mature barley seeds. Why? The authors should be further discuss.

Response 2: Thanks the reviewer for this content. The synthesis and accumulation of grain protein is highly dependent on the availability of nitrogen fertilizer, and it is difficult to understand the underlying control mechanisms of this. In the present study, we analyze the change of grain protein content and subproteome of seed protein in barley. The result showed that albumin and globulin are thus two subproteomes involved in GSP synthesis in grain seeds that are affected by nitrogen levels, a lot of differentially expressed proteins were observed. For example, GAPDH, Serpin-Z4, Z7 and so on. On the other hand, hordein fractions were affected N supply, we demonstrated the expression profiles of hordein and glutelin under low and high nitrogen conditions, and specific differentially expressed hordein proteins were identified. A total of eight protein spots representing five hordein proteins were differentially expressed in Yangsimai 3 and Naso Nijo, which could contribute to the difference in GPC of mature barley seeds at the protein level. 

Comment 3. The introduction should explain the aim of the study.

Response 3: The purpose of this study as follows: In the present study, we report a comparative proteomics analysis of two cultivars (Yangsimai 3 and Naso Nijo) in high and low nitrogen conditions using two-dimensional gel electrophoresis (2-DE) and tandem Mass Spectrometry (MS). The main objectives were: (1) to investigate the difference in GPC and protein composition between low and high nitrogen-treated plants; (2) to obtain comparative information on the subproteomes of feed barley and malting barley under low and high nitrogen conditions; (3) to identify changes in protein composition and candidate proteins that increase GPC under high nitrogen conditions.

Comment 4. References should be in uniform format as required by Plos one publication.

Response 4: According to the reviewer’s suggestion, we have revised the manuscript according to the PLOS ONE's style requirements.

Responses to reviewer 2

Comment 1. The MS is a bit too descriptive and lacks major conclusion. The mechanisms underline the changes need to be discussed, i.e. many differentially expressed proteins were detected (the same as previously reported) but their roles in causing the difference between two varieties were not discussed.

Response 1: Thanks the reviewer for this comment. In the present study, the hordein fraction in barley grains was more sensitive to nitrogen and increased as the nitrogen level increased, which is the most stable factor for high GPC in feed barley. In addition, eight differentially expressed protein spots were detected in hordein proteome profile, which could contribute to the difference in GPC of mature barley seeds on the protein level.

Comment 2. Not sure if it's appropriate to compare the results with previously reported QTL.

Response 2: Thanks the reviewer for this comment. We have deleted QTL analysis in the revised manuscript.

Comment 3. Can the authors provide soil test results (how much N in the soil before N application)?

Response 3: According to the reviewer’s suggestion. We test the total N in the soil. Both barley cultivars were planted at the Yangzhou University Experimental Farm in autumn of 2014 in soil with a total N of 1.14g/Kg

Comment 4. Don't think they can use “biological" before replicates if their replicates are from a single experiment.

Response 4: Thanks the reviewer for this comment. A total of 100 seeds per genotype used for protein extraction and protein content measurement for each replication. Three biological replicates were performed in the present study.

Comment 5. A conclusion paragraph is needed at the end of Discussion.

Response 5: According to the reviewer’s suggestion. A conclusion paragraph was added in the revised manuscript. 

Conclusion: In the present study, the GPC of mature grain was significantly higher in Yangsimai 3 than in Naso Nijo following nitrogen treatment. Hordein content was higher in both cultivars under high nitrogen conditions. Subproteome analysis revealed 152 differentially expressed protein spots on 2-DE gels. A total of 42 and 66 protein spots were successfully identified by tandem mass spectrometry in Yangsimai 3 and Naso Nijo under different nitrogen levels. Remarkably, the variation in B hordein content in Yangsimai 3 could have contributed to the higher GPC in Yangsimai 3 compared to Naso Nijo under different nitrogen availability conditions. Therefore, storage protein accumulation leads to GPC variation in feed and malting barley.

---

## [Decision Letter · Decision Letter 1]

1 Oct 2019

Regulation of nitrogen availability results in changes in grain protein content and grain storage subproteomes in barley (Hordeum vulgare L.)

PONE-D-19-19801R1

Dear Dr. Xu,

We are pleased to inform you that your manuscript has been judged scientifically suitable for publication and will be formally accepted for publication once it complies with all outstanding technical requirements.

With kind regards,

Chengdao Li, PhD

Academic Editor

PLOS ONE

Additional Editor Comments (optional):

The paper can be accepted for publication. However, there are still a few spelling and grammar errors, e.g. Hordeum not Hordeums. Please have a final read and correct all the errors along with any remaining formatting changes. 

Reviewers' comments:

Reviewer's Responses to Questions

**Comments to the Author**

1. If the authors have adequately addressed your comments raised in a previous round of review and you feel that this manuscript is now acceptable for publication, you may indicate that here to bypass the “Comments to the Author” section, enter your conflict of interest statement in the “Confidential to Editor” section, and submit your "Accept" recommendation.

Reviewer #2: All comments have been addressed

2. Is the manuscript technically sound, and do the data support the conclusions?

Reviewer #2: Yes

3. Has the statistical analysis been performed appropriately and rigorously? 

Reviewer #2: Yes

4. Have the authors made all data underlying the findings in their manuscript fully available?

Reviewer #2: Yes

5. Is the manuscript presented in an intelligible fashion and written in standard English?

Reviewer #2: Yes

6. Review Comments to the Author

Reviewer #2: I have no further comments apart from a few spelling errors, i.e. Hordeums vulgare L.should be Hordeum vulgare L.

7. PLOS authors have the option to publish the peer review history of their article (what does this mean?). If published, this will include your full peer review and any attached files.

Reviewer #2: No

---

## [Editor Report · Acceptance letter]

7 Oct 2019

PONE-D-19-19801R1 

Regulation of nitrogen availability results in changes in grain protein content and grain storage subproteomes in barley (*Hordeum vulgare* L.) 

Dear Dr. Xu:

I am pleased to inform you that your manuscript has been deemed suitable for publication in PLOS ONE. Congratulations! Your manuscript is now with our production department. 

With kind regards,

on behalf of

Professor Chengdao Li 

Academic Editor

PLOS ONE